# Smart pH- and Temperature-Sensitive Micelles Based on Chitosan Grafted with Fatty Acids to Increase the Efficiency and Selectivity of Doxorubicin and Its Adjuvant Regarding the Tumor Cells

**DOI:** 10.3390/pharmaceutics15041135

**Published:** 2023-04-03

**Authors:** Igor D. Zlotnikov, Dmitriy A. Streltsov, Alexander A. Ezhov, Elena V. Kudryashova

**Affiliations:** 1Faculty of Chemistry, Lomonosov Moscow State University, Leninskie Gory, 1/3, 119991 Moscow, Russia; zlotnikovid@my.msu.ru (I.D.Z.);; 2Faculty of Physics, Lomonosov Moscow State University, Leninskie Gory, 1/2, 119991 Moscow, Russia; alexander-ezhov@yandex.ru

**Keywords:** polymeric micelle, pH sensitive, anticancer selectivity, efflux

## Abstract

The main factors that determine the low effectiveness of chemotherapy are the low target bioavailability of antitumor drugs and the efflux process. In attempts to overcome this problem, several approaches are proposed here. Firstly, the development of polymeric micellar systems based on chitosan grafted by fatty acids (different types to optimize their properties), which, on the one hand, increase the solubility and bioavailability of cytostatics and, on the other hand, effectively interact with tumor cells due to the polycationic properties of chitosan, allowing for more effective penetration of cytostatic drugs into the cells. Secondly, the use of adjuvants—synergists of cytostatics (such as eugenol) included in the same micellar formulation—that selectively enhance the accumulation and retention of cytostatics in the tumor cells. pH- and temperature-sensitive polymeric micelles developed show high entrapment efficiency for both cytostatics and eugenol (EG) >60% and release the drug in a prolonged manner for 40 h in a weakly acidic medium corresponding to the microenvironment of tumors. In a slightly alkaline environment, the drug circulates longer (more than 60 h). The thermal sensitivity of micelles is realized due to an increase in the molecular mobility of chitosan, which undergoes a phase transition at 32–37 °C. The effect of the cytostatic drug doxorubicin (Dox) on cancerous A549 cells and model healthy cells of human embryonic renal epithelium (HEK293T) was studied by FTIR spectroscopy and fluorescence microscopy. Micellar Dox penetrates into cancer cells 2–3 times more efficiently when using EG adjuvant, which inhibits efflux, as demonstrated by a significant increase in the ratio of intra- and extracellular concentrations of the cytostatic. However, here it is worth remembering about healthy cells that they should not be damaged: according to changes in the FTIR and fluorescence spectra, the penetration of Dox into HEK293T when using micelles in combination with EG is reduced by 20–30% compared to a simple cytostatic. Thus, experimental developments of combined micellar cytostatic drugs have been proposed to increase the effectiveness of cancer treatment and overcome multiple drug resistance.

## 1. Introduction

Self-assembling supramolecular assemblies have a number of advantages as drug delivery systems: (1) amphiphilic molecules allow the dissolution of hydrophobic and hydrophilic drugs; (2) dynamically controlled formations; (3) beneficial for health (natural components of chitosan and fatty acids and antioxidants); (4) they interact with cell membranes to a greater extent with tumor cells due to the peculiarities of the morphology and structure of tumor cells; and (5) it is possible to create gels, ointments, and other convenient therapeutic forms [1,2,3].

Numerous self-assembling supramolecular drug carriers have been described in the literature [2,4,5,6,7]: polymer particles, liposomes, micelles, etc. There are significant changes in the effectiveness of anti-cancer drugs (doxorubicin, paclitaxel): drug delivery systems have shown significantly higher antitumor activity in vivo, enhanced the drug’s cellular uptake, and increased cells’ death. We proposed polymeric micelles due to their advantages: biocompatibility, high drug loading due to their amphiphilicity, in vivo efficiency, and, in comparison, with liposomes, their ease of preparation and higher thermodynamic stability. Furthermore, liposomes tend to carry drugs mostly to the liver, limiting their applicability. Polymer micelles, self-assembled colloidal systems composed from amphiphilic block copolymers, form a two-phase spherical structure in aqueous solutions with a hydrophobic inner core and a hydrophilic outer shell [1]. Such micelles have great potential in numerous applications, with their ease of production, regularity in size, and the presence of a core with apolar properties that provides hydrophobic drug solubilization. Polymeric micelles are nanoscale drug delivery systems, normally of the order of magnitude of 100 nm, which is suitable in terms of influencing both cell internalization and clearance from circulation. Indeed, literature guidance states that nanoparticles should be sized smaller than the sinusoids in the spleen and the fenestra of the liver (150–200 nm) to avoid clearance and that passive tumor targeting through the enhanced permeation and retention effect (EPR) occurs through gaps of 100–600 nm [8,9,10]. Polymer micelles, in comparison with simple micelles, have a much lower critical micelle concentration (CMC) and higher kinetic stability [11].

In this work, we propose an approach that increases the therapeutic potential of cytostatics through the use of polymeric pH- and thermo-sensitive micelles as well as the exploitation of the adjuvants (enhancers of cytostatics), since the main factors that determine the low effectiveness of chemotherapy are the low target bioavailability of antitumor drugs and the efflux process (release of drugs from cells by P-glycoprotein, multidrug-resistant proteins). Micelles are necessary for the dissolution of microcrystals of organic molecules due to the amphiphilic structure of copolymers, effective adsorption, concentration of particles on the surface of tumor cells, and increased permeability for cytostatics. The use of molecular containers for cytostatics will protect the drug from destruction, increase the duration of action, and reduce toxicity. Multiple drug resistance is directly caused by efflux pumps and low influx. In addition, adjuvants from plant extracts themselves have anticancer activity [12,13,14,15,16,17,18,19] and show synergism with cytostatics by inhibiting efflux (pumping the drug out of cells) and increasing the permeability of the membrane of cancer cells, and have prospects to show high selectivity relative to healthy due to the difference in morphology and molecular patterns of the cells of healthy and cancer. We have previously studied the process of inhibition of efflux pumps in bacterial cells of *E. coli* and *B. subtilis* by eugenol, menthol, apiol, and their analogs [19,20]. It is logical to assume the effectiveness of these substances in tumor cells due to the similarity of structural motifs with the inhibitors used in the literature. Advantages: action by an alternative mechanism compared to main drugs; biocompatibility of plant extracts; high efficiency; and minimal side effects.

Inflammation strongly manifests itself in the extracellular and intracellular pH profiles of the biological system. In solid tumors, within their extracellular microenvironment, pH is more acidic (~6.5) than in blood media (pH 7.4) [21]. In addition, the pH inside the tumor cells is also significantly lower (~5.5) compared to normal cells [22]. Therefore, pH-sensitive drug delivery systems selectively release cytostatics in tumors. pH-sensitive polymers ionize in response to changes in the acidity of the medium, undergo conformational changes, and release the drug or, conversely, preserve it [23]. The main approaches to the creation of pH-sensitive platforms are: (1) ionizable fragments cause conformational and structural changes in polymers; (2) pH-sensitive drug-polymer cross-links (such as Schiff bases) break with increasing acidity, resulting in drug release; and (3) ionization capacity (polyethyleneimine) [23].

The pH-sensitive release from micelles is due to: (1) protonation or deprotonation of polymers, leading to the destruction of the micellar structure; (2) reduced hydrophobicity of the hydrophobic segment of polymer micelles, which causes swelling of the micelles for drug release; and (3) breaking the acid–labile bond between drug and polymer [7].

A promising polymer for creating pH-sensitive micelles is chitosan, which has the properties of biocompatibility, biodegradability, mucoadhesivity, non-toxicity, antimicrobial activity, and antitumor activity [24,25,26,27,28,29,30,31,32,33,34,35,36,37,38,39,40,41,42,43,44,45]. Chitosan contains a large number of amino groups, which provide the polycationic properties of the polymer. This explains the ability of chitosan to bind and firmly hold a large amount of organic substances (including drugs that are poorly soluble in water). Only low-molecular weight chitosan is water soluble and, at the same time, has low immunogenicity. Chitosan-based micelles tend to release the drug at pH 5 [29], which causes the predominant drug release in tumors and endosomes. In addition, a chitosan-based delivery system can respond to temperature and release the drug at 37 °C due to the increased molecular mobility of chains. The second component of micelles is fatty acid residues (which varied in order to optimize the properties), which are natural substances that restore the membranes of healthy cells, exhibit an antioxidant effect (lipoic acid), and dissolve the drug.

Thus, the present work is aimed at developing experimental bases for creating effective forms of cytostatic drugs through the use of pH-sensitive drug release and efflux inhibition technology (cytostatics and adjuvants in soluble form in one molecular container).

## 2. Materials and Methods

### 2.1. Reagents

Chitosan oligosaccharide lactate 5 kDa, oleic acid (OA), stearic acid (SA), lipoic acid (LA), 11-mercaptoundecanoic acid (MUA), 1-Ethyl-3-(3-dimethylaminopropyl) carbodiimide (EDC), N-hydroxysuccinimide (NHS), 1 M 2,4,6-trinitrobenzenesulfonic acid (TNBS), and doxorubicin (Dox) hydrochloride were obtained from Sigma-Aldrich (St. Louis, MI, USA). Eosin-5-isothiocyanate was purchased from Invitrogen (Molecular Probes, Eugene, OR, USA). Eugenol of the highest commercial quality was purchased from Acros Organics (Flanders, Belgium). The preparation of apiol and plant extracts was carried out in the same way as described earlier [19]. 

### 2.2. Synthesis and Characterization of Micelles

#### 2.2.1. Synthesis of Grafted Chitosans and Modification Degree Determination

The chemical conjugates of Chit5-SA, Chit5-OA, Chit5-MUA, and Chit5-LA were synthesized by the coupling reaction of carboxyl groups of acids with amine groups in the presence of 1-ethyl-3-(3-dimethylaminopropyl) carbodiimide (EDC) at 60 °C for 12 h. The oleic acid (OA), stearic acid (SA), lipoic acid (LA), and 11-mercaptoundecanoic acid (MUA) (20 mg) were dissolved in 4 mL ethanol + 1 mL CH_3_CN. A 3-fold molar excess of EDC and a 2-fold molar excess of NHS were used. The reaction mixture was dialyzed against a 50% ethanol solution using a dialysis membrane (MWCO 6–8 kDa) for 12 h and then against water for 12 h.

All samples were freeze-dried at −60 °C (Edwards 5, BOC Edwards, Burgess Hill, UK). The degree of modification was calculated according to the spectrophotometric titration of amino groups (before and after modification) with 2,4,6-trinitrobenzenesulfonic acid (TNBS) using a 1 M TNBS solution in a 1 M sodium-borate buffer (pH 9.2). The content of amino groups in unmodified chitosan was assumed to be 100%.

Eosin-labelled micelles were obtained by conjugation reaction: chitosan NH_2_-group with activated eosin. The micelle’s samples were incubated for 60 min at 37 °C (PBS, pH 7.4) with 1 µg/mL of eosin-5-isothiocyanate solution followed by dialysis against PBS (cut-off 6–8 kDa) for 4 h.

#### 2.2.2. Preparation of Micelles—Critical Micelle Concentration (CMC)

Polymers were dissolved in PBS (0.01 M, pH 7.4) at a concentration of 2 mg/mL. Micelle solutions were prepared by probe-type ultrasonic treatment (50 °C, 10 min). The CMC was determined using Nile Red dye. Nile Red was dissolved in DMSO (1 mg/mL), followed by a ×1000 (×100) dilution in PBS. Fluorescence emission spectra depend on the molar excess of micelles, where the analytical signal (fluorescence intensity) correlates with the formation of polymeric micelles. CMC was determined based on the coordinates of the point at the half-height of the hyperbola (intensity as a function of polymer concentration). PBS (0.01 M, pH 7.4). λ_exci_ = 490 nm. 

The spectrophotometric determination of CMC was performed using deconvolution of the absorption spectra of Nile red in its free state and micellar formulation into hydrophilic and hydrophobic components with Gaussian. The steep slope of the sigmoid (fraction of the hydrophobic component of the spectrum, corresponding to the dye inside the micelles, as a function of the concentration of the micelles) corresponds to the formation of micelles. The inflection point corresponds to the CMC. 

#### 2.2.3. Doxorubicin Loading into Micelles

Doxorubicin (1 mg/mL) was mixed with copolymer micellar solutions (1 mg/mL), and then suspensions were sonicated for 30 min at 50 °C. For drug loading capacity determination, analytical dialysis against distilled water for 12 h at 37 °C was performed using a 6–8 kDa cut-off dialysis membrane with a 1:10 internal-to-external volume ratio. Then the amount of Dox (according to A488) in the external solution and in the micelles were determined.

Additionally, the fraction of incorporated Dox in a micelle was determined from the deconvolution of the FTIR spectra into 3 components: the hydrophobic microenvironment in the core of micelles (1583–1586 cm^−1^), Dox in solution (1572–1574 cm^−1^), and the hydrophilic microenvironment in chitosan chains (1563–1566 cm^−1^).

Equations [19]:Entrapment efficiency of Dox: EE (%) = 100 · (Dox amount in micelle)/(total Dox amount)(1)
Loading capacity of Dox: LC (%) = 100 · (mass of loaded Dox into micelles)/(mass of sample)(2)

#### 2.2.4. Doxorubicin Release from Micelles

Samples from Section 2.2.3 were freeze-dried as described above. Doxorubicin samples (free and micellar) are dissolved in PBS (pH = 7.4, 0.01 M) or sodium acetate buffer (pH = 5.5, 0.01 M) to a drug concentration of 1 mg/mL. Release of Dox from micelles was performed using a dialysis membrane (cut-off, 12–14 kDa) to an external 10 mL PBS buffer solution at 37 °C. The amount of Dox was determined by absorption at 488 nm and fluorescence intensity. Absorption spectra of solutions were recorded on the AmerSham Biosciences UltraSpec 2100 Pro device (Woburn, MA, USA) in the range of 400–600 nm. The fluorescence of Dox was measured using a Varian Cary Eclipse spectrofluorometer (Agilent Technologies, Santa Clara, CA, USA) at 22 °C: λ_exci_ = 488 nm, λ_emi_ = 560 nm.

#### 2.2.5. Determination of the Hydrodynamic Diameter of the Micellar Particles

Determination of the hydrodynamic diameter of the synthesized polymeric micelles was carried out by nanoparticle tracking analysis using the Nanosight LM10-HS device (Salisbury, UK). Particle samples were diluted with MilliQ-purified water (Merck Millipore, Burlington, MA, USA) to a particle concentration of 10^9^–10^10^ particles/mL. The hydrodynamic diameter was determined by the Stokes-Einstein equation due to the analysis of the trajectory of Brownian motion of particles. Each sample was measured five times. The results are averaged and presented with a standard deviation.

#### 2.2.6. Fluorescent Micelle Visualization

Fluorescent images of micelle’s particles containing doxorubicin were obtained using the ZOE Fluorescent Cell Imager (Bio-Rad Laboratories, Hercules, CA, USA).

### 2.3. Cell Cultivation and Toxicity Assay

Adenocarcinomic human alveolar basal epithelial cells (A549) cell lines (Manassas, VA, USA) were grown in an RPMI-1640 medium (Gibco, Thermo Fisher Scientific Inc., Waltham, MA, USA), and supplemented with 5% fetal bovine serum (Capricorn Scientific, Ebsdorfergrund, Germany) and 1% sodium pyruvate (Paneco, Moscow, Russia) at 5% CO_2_/95% air in a humidified atmosphere at 37 °C. 

Linear cells of the embryonic human kidney epithelium (HEK293T) are cultured in DMEM medium with 4.5 g D-glucose (Life Technologies, Carlsbad, CA, USA) supplemented with 10% fetal bovine serum (FBS) (Gibco, Waltham, MA, USA) and 100 units/mL of penicillin and streptomycin. Cell passaging occurs upon reaching 70–90% confluent monolayers. The following conditions are maintained in the incubator: temperature of 37 °C and 5% CO_2_ in the air at constant humidity. Removal of cells from culture plastic is carried out using a 0.05% trypsin/EDTA solution (Hyclone, Logan, UT, USA).

### 2.4. FTIR Spectroscopy Studying of Dox and Adjuvant Actions on A549 and HEK293T Cells

ATR-FTIR spectra of cell samples in suspension were recorded using a Bruker Tensor 27 spectrometer equipped with a liquid nitrogen-cooled MCT (mercury, cadmium, and telluride) detector. Samples were placed in a thermostatic cell, BioATR-II, with a ZnSe ATR element (Bruker, Bremen, Germany). FTIR spectra were acquired from 850 to 4000 cm^−1^ with 1 cm^−1^ spectral resolution. For each spectrum, 50 scans were accumulated and averaged. Spectral data were processed using the Bruker software system Opus 8.2.28 (Bruker, Bremen, Germany).

A549 or HEK293T cells’ suspensions (1.5 × 10^6^ cells/mL) were washed twice with sterile PBS (pH = 7.4) from the culture medium by centrifuging (Eppendorf centrifuge 5415C, 10 min, 3000× *g*). The cells are precipitated by centrifugation and separated from the supernatant, washed twice, and resuspended in PBS (5 × 10^6^ cells/mL) to register FTIR spectra. 

Cell suspensions were incubated with Dox-containing samples, and FTIR spectra were registered at 37 °C online or after 0.5–1–2–3 h of incubation. To quantify absorbed Dox, the cells were precipitated by centrifugation and separated from the supernatant, washed twice, and resuspended in 50 µL PBS to register FTIR spectra. The supernatant was separated to determine the amounts of unabsorbed substances.

### 2.5. Fluorescence Microscopy of Cells

Fluorescence images of cells were obtained by an inverted microscope, the Olympus IX81, equipped with an Olympus XM10 cooled CCD monochrome camera. A xenon arc lamp was used as a light source for fluorescence imaging, and a halogen lamp was used for transmitted light imaging. The excitation and emission wavelength ranges were selected by Olympus U-MNB2 and U-MWG2 fluorescence mirror units for blue and green excitation light, respectively. Dry Olympus objectives UPlanSApo 20X NA 0.75 and UPlanSApo 40X NA 0.90 were used for the measurements. Transmitted light images were obtained by the DIC technique. The Olympus Cell Sens imaging software v.3.2 was used for microscope and camera control. Obtained images were treated by the ImageJ 1.53e software.

### 2.6. Atomic Force Microscopy (AFM)

Topography, phase, and magnitude signal images of the micelles deposited onto a freshly cleaved mica surface were obtained by atomic force microscopy (AFM) using a scanning probe microscope, NTEGRA Prima (NT-MDT, Moscow, Russia), operated in a semi-contact mode with a 15–20 nm peak-to-peak amplitude of the “free air” probe oscillations. Silicon cantilevers NSG01 “Golden” series cantilevers for semi-contact mode (NT-MDT, Russia) were used. Image processing was performed using the Image Analysis software (NT-MDT, Russia).

### 2.7. NMR Spectroscopy

An amount of 5–10 mg of the sample was dissolved in 600 μL of D_2_O. ^1^H-spectra of the solutions were recorded on a Bruker Avance 400 spectrometer (Germany) with an operating frequency of 400 MHz.

### 2.8. Cell Cultivation and Toxicity Assay

Adenocarcinomic human alveolar basal epithelial cells A549 cell lines (Manassas, VA, USA) were grown in RPMI-1640 medium (Gibco, Thermo Fisher Scientific Inc., Waltham, MA, USA) supplemented with 5% fetal bovine serum (Capricorn Scientific, Ebsdorfergrund, Germany) and 1% sodium pyruvate (Paneco, Moscow, Russia) at 5% CO_2_/95% air in a humidified atmosphere at 37 °C. Cell lines were tested for mycoplasma contamination before the experiment using the Mycoplasma Detection Kit PlasmoTest™ (InvivoGen, San Diego, CA, USA).

Cell lines were obtained from the Laboratory of Medical Biotechnology, Institute of Biomedical Chemistry (Moscow, Russia).

To test acute toxicity, cells were cultivated for 72 h in a V-bottom 96-well plate (TPP, Trasadingen, Switzerland) in the presence of Pac, Dox, cisplatin, and their combined formulations, and cell viability was tested by measuring the conversion of the tetrazolium salt, 3-(4,5-dimethyl-thiazol-2-yl)-2,5-diphenyltetrazolium bromide (Serva, Heidelberg, Germany), to formazan (MTT test) [46,47].

### 2.9. Statistical Analysis

A statistical analysis of the obtained data was carried out using Student’s *t*-test Origin 2022 software (OriginLab Corporation, Northampton, MA, USA). Values are presented as the mean ± SD of three experiments (three replicates).

## 3. Results

### 3.1. Synthesis and Characterization of Polymeric Micelles 

The synthesis of chitosan 5 kDa (Chit5) grafted with fatty acids was carried out by activating the acid carboxyl group by EDC and NHS to form a stable intermediate (acid residue–NHS), followed by conjugation with amino groups in Chit5, as shown in Figure 1a. The set of different fatty acids (chain length, the presence of double bonds, the presence of SH groups in the form of S-S or SH) was varied to obtain micelles with different physico-chemical properties (size, critical micelle concentration (CMC), drug release rate, loading efficiency, and effect on cells). The fluorescent image of Chit5-OA-20 micelles, labeled with Dox, is shown in Figure 1b; the photos show particles smaller than a micron—aggregates of micelles with a dye included in the core. The formation of micelles is accompanied by the loading of the drug, so we observe fluorescent particles and their aggregates. The morphology of Chit5 polymer aggregates and Chit5-MUA-20 micelles was studied using AFM (Figure 1c,d). As shown in Figure 1c, the chitosan polymer forms aggregated particles with a high degree of heterogeneity in structure and size, ranging from 10 to 100 nm. At the same time, Chit5-MUA-20 micelles turned out to be homogeneous, spherical, non-aggregated particles with an average size of 30 nm (Figure 1d). Furthermore, judging by the qualitative difference between the phase images of the particles in the non-micellar (polymeric) sample and the micellar particles sample, it can be assumed that these particles are different in their textural properties. The micellar sample has a microphase inside the particles that provides other mechanical properties (hydrophobic phase in micelles).

The chemical structure was studied using FTIR and ^1^H NMR spectroscopy (Figure 2). In the FTIR spectra of chitosan–acid conjugates, broad, intense bands appear (in comparison with Chit5), corresponding to the O–H and N–H stretching oscillations, which correspond mainly to the bonds involved in the formation of hydrogen bonds [43,48,49], which is typical of micellar systems. The FTIR spectrum of oleic acid was presented earlier in the work [2]: the characteristic peaks at 2982, 2925, and 2857 cm^−1^ of the C-H stretching oscillations are shifted during conjugation (acid vs. chitosan-acid) to 2960–2965, 2913–2917, and 2848–2850 cm^−1^, respectively, which indicates a dense packing of lipids in polymer structure, which corresponds to a transition to a more structured state with a dense lipid package just as in liposomes [50].

The conjugate’s formation of acid residues on chitosan was confirmed by shifts of the peaks of the COOH group (1730–1770 cm^−1^) of fatty acids to the low-frequency region due to the formation of amide bonds: 1500–1600 cm^−1^ (amide –NH–) and 1640–1720 cm^−1^ (–C(=O)–). Bands of carbonyl groups (Figure 2a, 1700–1750 cm^−1^) are multicomponent and sensitive to changes in the degree of hydration. The formation of micelles is accompanied by a decrease in the degree of hydration and increases during the phase transition. For the considered grafted chitosans, the shift of the C=O band to the high-frequency region and the decrease in intensity of the CH_2_ groups of acid residues (Figure 2a) confirm the formation of micellars (at concentrations above CMC).

^1^H NMR spectra of Chit5 and Chit5 grafted with acids are presented in Figure 2b–d. In the ^1^H NMR spectra of Chit5, characteristic peaks (δ, ppm) are observed [24,30,38,43,51]: 4.22 (H1), δ = 3.23 (H2), δ = 3.79, 3.96 (H3, H4, H5, H6, H6′), δ = 2.11 (NH–C(=O)–CH_3_). ^1^H NMR spectra of the Chit5-MUA-20 and Chit5-LA-20 (Figure 2c,d) contain both signals of two components (chitosan and acid): 1.41 (S–H of MUA), increased signals at 2.0–2.3, and 1.25 were assigned to the methene hydrogen of the N-alkyl groups [52,53,54]. Signals of 3.64 ppm (C–H near the dithiolane fragment) and 2.3 ppm (β–H with relation to the carboxyl group) indicate the presence of LA in the conjugate [54]. Thus, NMR spectra confirm the structures of compounds described by the FTIR method.

Based on FTIR spectroscopy data (integral peak intensities), nanoparticle tracking analysis, TNBS amino-group titration (Figure 3), and initial component ratios, the average modification degrees and chemical composition of conjugates were determined (Table 1). The degree of acid grafting of chitosan varied from 12 to 20% (the theoretical maximum possible degree of modification of amino groups according to the method is 20%). This corresponds to the modification of one chitosan molecule with 3–7 residues of oleic, stearic, 11-mercaptoundecanoic, or lipoic acids. Depending on the acid residue, polymer micelles will have different properties: hydrodynamic size, packing density, drug interaction, and CMC. However, in general, all polymers turned out to comply with the standard in terms of micelles formation. According to the literature data, an increase in the degree of chitosan modification to a certain value (up to 30–40%) [2,11,28,29,30,41,43,44,55] improves micelle formation due to the compaction of the hydrophobic core and, consequently, the thermodynamically more preferable inclusion of aromatic drugs. A small degree of modification causes a loose core and a larger micelle size. With a decrease in the length of the fatty acid residue, the size of micelles decreases, however, in the case of MUA, disulfide bonds inside micelles can form, increasing the size of micelles (Table 1).

### 3.2. Critical Micelle Concentration (CMC)

The determination of CMC can be carried out most accurately using fluorescence techniques. In a previous work [2], we optimized the methodology using pyrene conjugated with chitosan. The technique presented here is based on using a non-covalent Nile Red label, the position of the maximum and the fluorescence intensity of which change upon transition to a hydrophobic microenvironment. Figure 4a shows the Nile Red fluorescence emission spectra depending on the type and excess of polymers forming micelles. The formation of micelles is accompanied by the incorporation of hydrophobic Nile red into the core of the micelles from the DMSO/H_2_O (1/1000 *v*/*v*) microenvironment. Indeed, this is reflected in the fluorescence emission spectra (Figure 4a): an intensity decrease and broadening of the peak with a shift from 520 nm to 525–530 nm is observed. The dependences of the fluorescence intensity on the concentration of micelle-forming polymers are presented in conventional and semi-logarithmic coordinates (for the convenience of curve analysis) in Figure 4b. A 50% change in fluorescence from the maximum change corresponds approximately to the CMC. Additionally, the control determination of CMC was also carried out by spectrophotometry, but in this case, it was necessary to have 1–2 orders of magnitude higher dye concentration. Figure 4c shows the absorption spectra of Nile Red in simple form and in micellar form. The spectrum consists of two components: hydrophilic (green) and hydrophobic (pink). The share of the latter increases when the dye is included in micelles. Similarly to the fluorescence method, we determine CMC values, which are presented in Table 1. Note that the values are of the order of 10^−8^–10^−9^ M, and CMC is 3–5 orders of magnitude lower when single lipids and surfactants are used. At the same time, the lowest CMC values are achieved for OA- and LA-grafted chitosans. Previously, it was shown that the CMC decreased from approximately 30 to 4 nM with an increase in the degree of modification of chitosan with oleic acid from 5 to 30% due to compactization [2]. Comparing techniques with the use of pyrene-labeled chitosan and non-covalent inclusion of Nile Red label into micelles, it should be noted that the pyrene-based technique turned out to be more laborious due to the production of covalent conjugates but more sensitive and convenient in analytical terms.

### 3.3. Loading of Doxorubicin into Micelles—Properties of Micellar Formulations

The spontaneous inclusion of cytostatics into the micellar system is one of the key issues of the article. FTIR spectroscopy provides valuable information about the bonds of atoms and the microenvironment of molecules, so the method is sensitive to intermolecular interactions as well as micelle formation. The “aromatic” fragments (1565–1595 cm^−1^) of the FTIR spectra of doxorubicin are shown in Figure 5a–c. The peaks can be decomposed into three Lorentz components corresponding to hydrophilic (1565–1566 cm^−1^), hydrophobic (1582–1587 cm^−1^) and intermediate (1571–1578 cm^−1^) microenvironments of the aromatic cytostatic system. Dox interacts with chitosan polymer chains or is incorporated into the micelle core, which corresponds to an increase in the integral fraction of the corresponding component. Based on the proportions of components in the FTIR spectra, the entrapment efficiency parameters of doxorubicin in micelles were calculated (Table 2). High values are characteristic of all types of micelles, except those modified with stearic acid, for which a low degree of chitosan modification was obtained (due to a more inflexible fat chain and being relatively refractory). At pH 7.4, the entrapment efficiency value is 40–50% (Table 2, Figure 5c): Dox is incorporated into the core of micelles, and part of it remains free or interacts with more hydrophilic chitosan. At the same time, in a slightly acidic medium (pH 5.5), the proportion of the hydrophilic chitosan-associated component increases (Figure 5b), that is, chitosan more effectively solvates the drug. Thus, at pH 5.5, entrapment efficiency reaches 60–70% (the chitosan component and the component in the core of micelles). In an acidic environment, chitosan is more soluble, micelles are looser, and Dox can enter but also come out faster.

### 3.4. pH-Sensitive and Thermosensitive Doxorubicin Release from Polymeric Micelle

As discussed in the introduction, the use of pH- and thermosensitive polymers is an important aspect of the targeted delivery of drugs to tumors. Dox has an amino group, which is protonated at low pH (pKa = 8.4). If Dox and chitosan (pKa 6–6.5) are both protonated in a slightly acidic environment, then the release is more active due to electrostatic repulsion. Therefore, micelles have the property of pH sensitivity (Figure 6, Table 3): at pH 5.5, the rate of Dox release is 2–3 times higher than at pH 7.4. Micelles are thermosensitive due to undergoing a phase transition and increasing the mobility of chitosan chains: the rate of Dox release increases 1.5–2 times with an increase in temperature from 25 °C to the physiologically relevant 37 °C, and 2–3 times when the temperature rises to 42 °C—the model temperature for the local inflammatory area (tumor microenvironment). Similar experiments were carried out for chitosan-coated liposomes the release of Dox at pH 5.5 occurs in half after 15 h, which is much longer than for the chitosan-based micelles under consideration, while at pH 7.4, liposomes are released a little slower by 20–30%, but the effect is not as bright as in the case of micelles. In addition to their advantages, the micelles are a simpler system than liposomes and do not require the use of organic solvents such as chloroform and methanol, which are difficult to completely remove. Further, the scalability of polymer-coated liposome production is difficult. Thus, we have shown that polymer micelles based on chitosan modified with fatty acids release the drug in an accelerated and controlled manner at pH 5.5 and 37–42 °C, which corresponds to the microenvironment of tumors or inflammatory foci.

The thermosensitivity of polymer micelles is the second aspect of the delivery system that can cause the targeted release of the drug. Figure 7a shows the FTIR spectra of Dox in Chit5-MUA-20 micelles at 22–45 °C, a phase transition of the micelles in which the drug is released. With an increase in temperature, the intensity of all peaks increases, especially those corresponding to fluctuations of N–H and O–H. In addition, a shift of the broad peak from 3400 cm^−1^ to 3600 cm^−1^ occurs, which indicates the rupture of intermolecular interactions (hydrogen bonds between chitosan–chitosan and chitosan–Dox) and the readiness of the micelle to release the drug. Significant changes occur in peaks at 1630 and 1080–1120 cm^−1^, corresponding to fluctuations in the C=O (Dox) and C–O–C bonds (chitosan and Dox) (Figure 7b). There is a consistent shift of the peak from 1643 to 1632 cm^−1^ and an increase in the proportion of the long-wave component in the complex peak of the C–O–C bond, which indicates a local decrease in the hydrophobicity of the Dox environment—one of the mechanisms of drug release from micelles. Thus, pH- and temperature-sensitive polymers based on chitosan have experimental prerequisites for targeted drug release in tumors.

### 3.5. FTIR Spectroscopy of Cancer and Normal Cells—Drug Interaction’s Tracking

The interaction of micellar systems with cells must be detailed at the molecular level in order to study the mechanism of enhancement of cytostatics. It is interesting how polymer micelles will manifest themselves when exposed to tumors and healthy cells. With the help of FTIR spectroscopy, it is possible to study the molecular details of the interaction of drugs with healthy and cancerous cells. In the cell, it is possible to distinguish the main structural units that contribute to the absorption in the IR region (Figure 8, Figure 9 and Figure 10): cell membrane lipids (2800–3000 cm^−1^), proteins, especially transmembrane (1500–1700 cm^−1^), DNA phosphate groups (1240 cm^−1^), carbohydrates, including lipopolysaccharides (900–1100 cm^−1^). The position of the peaks, shape, and intensity are sensitive to the binding of the bilayer with ligands or drug molecules, the formation of hydrogen bonds, aggregation, etc.

Figure 8 shows the FTIR spectra of cancer A549 cells and healthy HEK293T cells after 3 h of incubation with Dox-containing formulations, minus the initial spectra of cells + drugs, which allows you to track the changes in the cells. Apparently, free Dox penetrated into cancer cells, even worse than in healthy cells (by a peak intensity of 1650 cm^−1^), while micelles increased the accumulation of Dox in cancer cells, probably due to adsorption and partial internalization of particles. The greatest effect is achieved when using micellar Dox in combination with an efflux inhibitor (eugenol, EG): the accumulation of Dox is increased by more than three times. However, it is important to know how the formula acts on healthy cells. It turned out that our proposed micelle + EG system manifests protective properties (presumably, strengthening of the membrane and influence on ion channels): the penetration of Dox into HEK293T cells is reduced by 20%.

To explain the effect of micelles on reduced Dox permeability in HEK293T healthy cells, the authors decided to find out how temperature affects the interaction of micelles with cells (Figure 9). It turned out that the penetration of free Dox does not depend much on temperature, but micellar Dox penetrates only at low temperatures (there are no changes in the spectrum or insignificant changes at 37 °C, but much larger changes at 22 °C), which is explained by the thermo-sensitive properties of micelles described above. At human body temperature, micelles do not release Dox into healthy cells but rather protect and refine the membrane of healthy cells.

To detect detailed changes in the A549 (and HEK293T) FTIR spectra during the interaction of cells with Dox, an online scan was performed (for 45 min with an interval of 5 min) (Figure 10). In the case of micellar Dox, the difference in peak intensities increases twice, especially for Amide I (1650 cm^−1^) and the peak of DNA phosphate groups (1240 cm^−1^), which corresponds to a more effective penetration of Dox and intercalation of DNA. For cancer cells, a change in temperature from 37 to 22 °C reduces the effectiveness of micellar Dox; however, penetration into A549 cells in comparison with a simple Dox still remains almost 2 times more powerful. In the case of healthy cells, significant smaller changes in the FTIR spectra are observed when using micellar Dox with the addition of EG in comparison with simple Dox, which indicates the protective effect of Micelle + EG on the cell membrane.

### 3.6. Fluorescence Microscopy of A549 and HEK293T—Drug Interaction’s Visualization

To find out the mechanisms of action of polymer micelles and adjuvants, for example, a study of the interaction of drugs (Dox) with cells using fluorescence microscopy was conducted. 

Figure 11 shows fluorescent and optical images of cancer A549 cells in the Dox red channel. The uptake of Dox was studied depending on the composition of the formulation: Dox free; Dox in micelles of Chit5-MUA-20; Dox + EG; Dox in micelles of Chit5-MUA-20 + EG. The cells were incubated for 2 h; during this time, the efflux process was actively developing [20]. The intensities of cell-associated fluorescence characterize the efficiency of the accumulation of cells inside cancer cells and are qualitatively reflected in Figure 11 and numerically in Table 4 (pixel integration). Figure 11c is a fluorescent image of A549 cells incubated with Dox in eosin-labeled micelles: Dox’s background is subtracted, so only the distribution of polymers is visible. Figure 11c demonstrated that polymeric micelles of grafted chitosan are efficiently adsorbed on the surface of cells and partially penetrate inside through membrane fusion. A comparison of fluorescent images of cells incubated with micelles labeled with eosin and with loaded Dox in the Dox red and eosin magenta channels (Figure 11b,c) shows that micelles penetrate into cells, colocalize with Dox, and increase the Dox influx.

The difference in the absorption of the Dox by A549 cells is visible even in a simple microscope when using a higher concentration of cytostatic (Figure 11f,g): Dox with micelles selectively penetrates into cells, staining mainly them, and simple Dox, due to efflux and weak infusion, is contained in large quantities in the extracellular medium.

Figure 12 shows fluorescent images of normal HEK293T cells in the Dox green channel (green for ease of perception and so as not to confuse healthy cells with cancerous ones). The data of fluorescence microscopy correlate with the data of FTIR spectroscopy on the penetration of Dox-containing formulations in different types of cells: micellar Dox with enhanced EG penetrates cancer cells 2–3 times better than simple Dox, while in healthy cells the accumulation of micellar Dox with EG is reduced compared to simple Dox (Figure 12a).

For quantitative evaluation, the intensity of cell-associated fluorescence (Table 4), calculated from the analysis of fluorescent images and quantitative data on the fluorescence of unabsorbed Dox, increases for micellar Dox and with the addition of EG by about 30–40%, and when using micellar Dox in combination with the amplifier effect, more than 140%. The efflux is characterized by the ratio of A549-associated to background fluorescence (which correlates with the ratio of intra- and extracellular concentrations of the cytostatic). The larger this ratio, the smaller the efflux. When using EG, the distribution of cytostatics into cancer cells increases by more than three times. Based on the data in Table 3, it follows that EG + micelles are an effective formulation not only against cancer cells but also perform a protective function in healthy HEK293T cells (Dox accumulation is halved). Thus, the developed micellar systems with an adjuvant to enhance cytostatics open up new ways to solve the problem of oncological diseases and the multidrug resistance of tumors.

### 3.7. MTT Assay of Dox Anti-A549 Activity

The changes observed in the FTIR spectra of cells and visually using a microscope indicate an increase in the effectiveness of cytostatics against cancer cells; however, the MTT assay is the control method that quantitatively describes survival. Figure 13 shows the curves of cell survival dependence on the concentration of a Dox-containing formulation. The MTT assay data correlate with the above data obtained by other methods: indeed, the micellar form of Dox, enhanced with an adjuvant EG, increases the effectiveness of doxorubicin and decreases the IC50 from 600 to 15 nM.

## 4. Conclusions

The present work is aimed at establishing the possible approaches for overcoming multidrug resistance of tumor cells (which may be due to efflux) and for effective cancer treatment by increasing the efficiency and, most importantly, the selectivity of the action of cytostatics through the use of smart micellar systems and the use of adjuvants as cytostatic enhancers. Synthesized polymer micelles are based on chitosan and fatty acids; the first provides pH, thermal sensitivity, and increased interaction with cancer cells. Fatty acids are natural components that restore the membranes of healthy cells in combination with eugenol (EG), a component of essential oils. Polymeric micelles accelerate the release of the cytostatic at pH 5.5 by 80% in 40 h, while at pH 7.4 the drug circulates longer. Micelles undergo a phase transition at 32–37 °C, which accompanies the release of the drug. A new technique has been developed for the diagnosis and elucidation of the mechanisms of drug interaction with cells at the molecular level (FTIR spectroscopy) and is now actively expanding its application. Using FTIR spectroscopy and fluorescence microscopy, it was shown that micellar Dox in combination with an EG adjuvant (efflux inhibitor) penetrates A549 tumor cells 2–3 times more efficiently, while accumulation in healthy HEK293T cells is reduced by ~2 times. In other words, a cytostatic formulation selective for cancer cells has been obtained. This opens up prospects for the creation of drugs for effective cancer treatment.

## Figures and Tables

**Figure 1 pharmaceutics-15-01135-f001:**
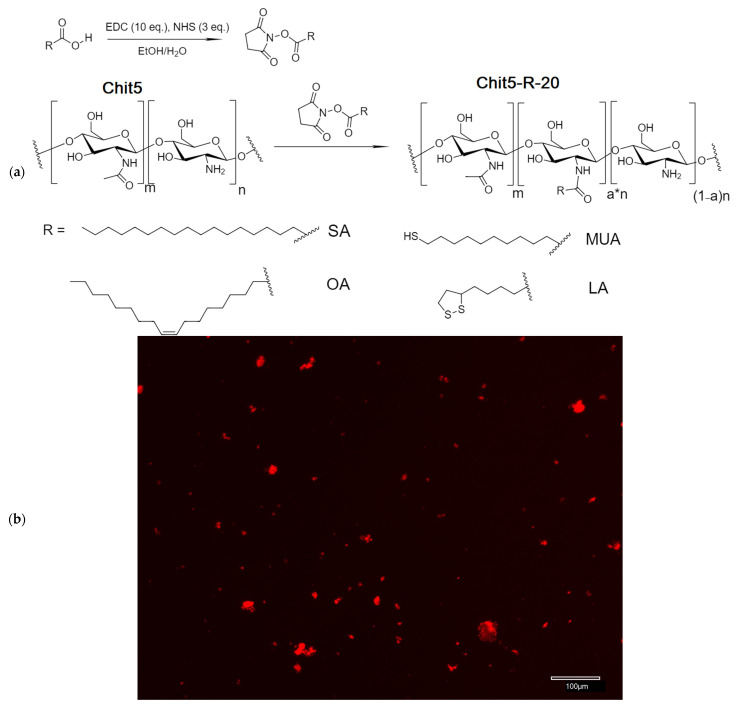
(**a**) The scheme of synthesis of grafted chitosans. m = 5—percentage of acylated pyranose fragments, n = 95—percentage of deacylated pyranose fragments, and a—experimental modification degree. (**b**) Fluorescent image of Chit5-OA-20 micelles and their aggregates, labeled with doxorubicin. (**c**) Atomic force microscopy images of Chit5. (**d**) Atomic force microscopy images of Chit5-MUA-20. (**e**) Magnitude signal image of the same area as (**d**).

**Figure 2 pharmaceutics-15-01135-f002:**
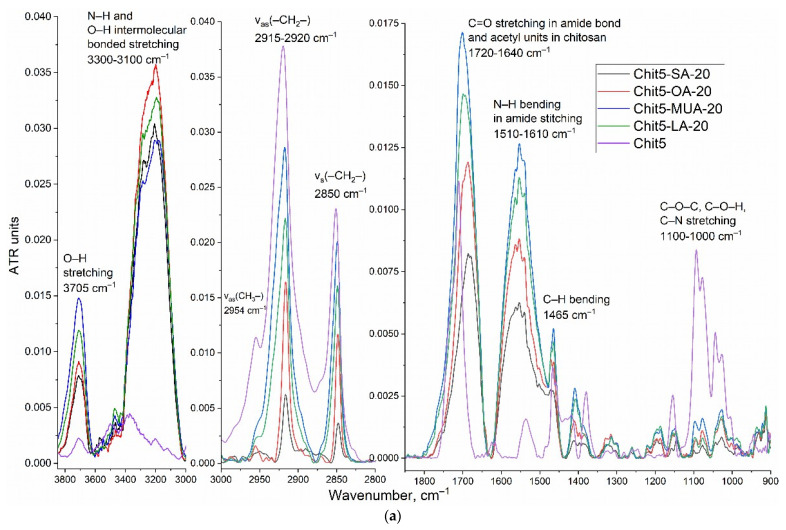
(**a**) FTIR spectra and (**b**–**d**) ^1^H NMR of Chit5 and Chit5 grafted with acid residues. PBS (0.01 M, pH = 7.4). T = 22 °C.

**Figure 3 pharmaceutics-15-01135-f003:**
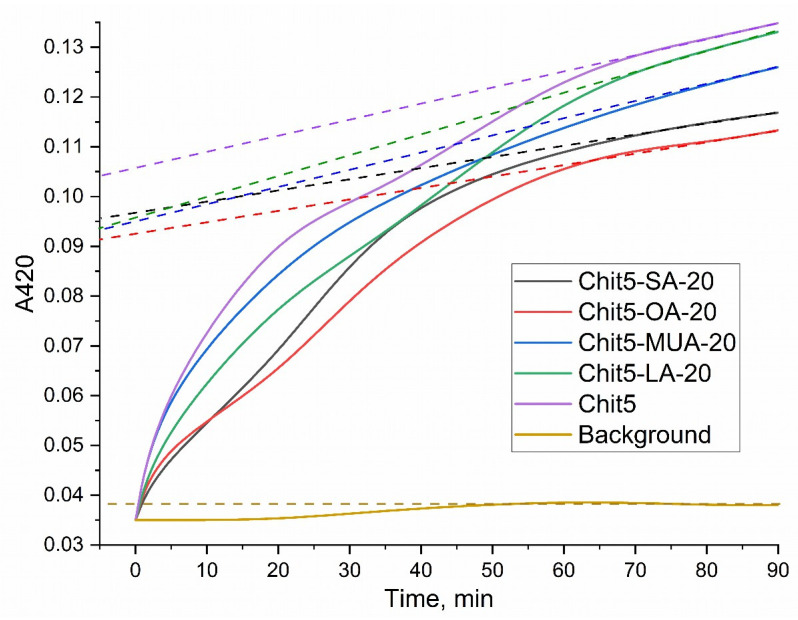
TNBS (2,4,6-trinitrobenzenesulfonic acid) spectrophotometric titration curves of Chit5 and its grafted derivatives. Tangents to curves (dashed lines) drawn to the ordinate axis give the value A420, from which the number of primary amino groups was calculated. T = 22 °C. 0.02 M Na_2_B_4_O_7_ (pH 9.2).

**Figure 4 pharmaceutics-15-01135-f004:**
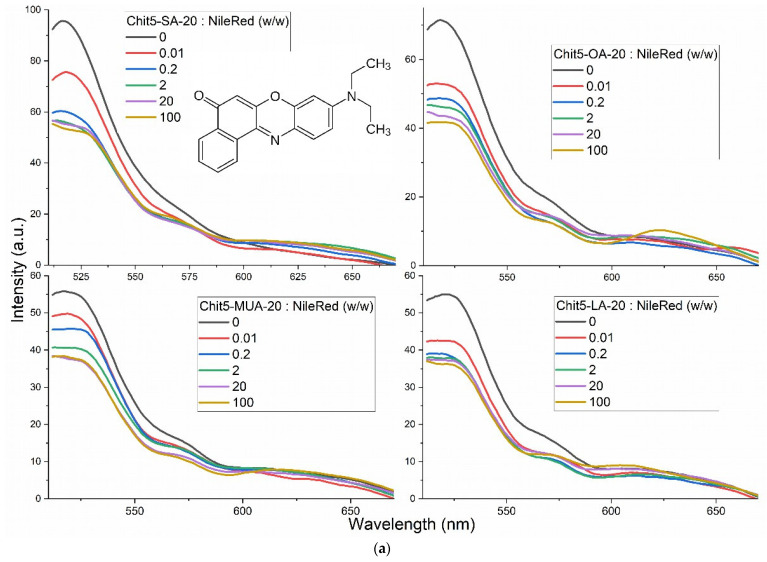
Fluorescence and spectrophotometric determination of the critical micelle concentration with Nile Red dye. (**a**) Fluorescence emission spectra of Nile red, free or with micelles. (**b**) Peak intensities of the spectra (**a**) dependences on the concentration of polymers. λ_exci_ = 490 nm. (**c**) Absorption spectra of Nile red in free state and micellar formulation with corresponding deconvolution into hydrophilic and hydrophobic components with Gaussians. PBS (0.01 M, pH 7.4). Nile Red was dissolved in DMSO (1 mg/mL), followed by ×1000 (or ×100) dilution in PBS.

**Figure 5 pharmaceutics-15-01135-f005:**
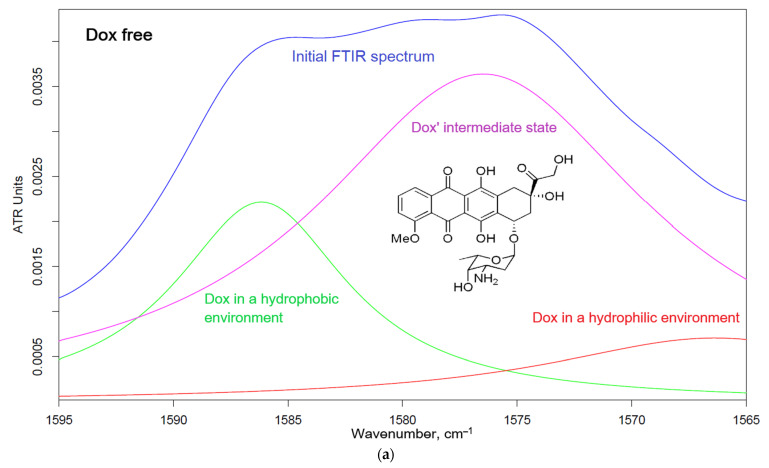
Doxorubicin FTIR spectra («aromatic» region) deconvolution with Lorentzians: (**a**) in free form, (**b**) loaded into Chit5-OA-20 micelles at pH = 5.5 (sodium acetate buffer, 0.01 M), and (**c**) loaded into Chit5-MUA-20 micelles at pH = 7.4 (PBS, 0.01 M). T = 37 °C.

**Figure 6 pharmaceutics-15-01135-f006:**
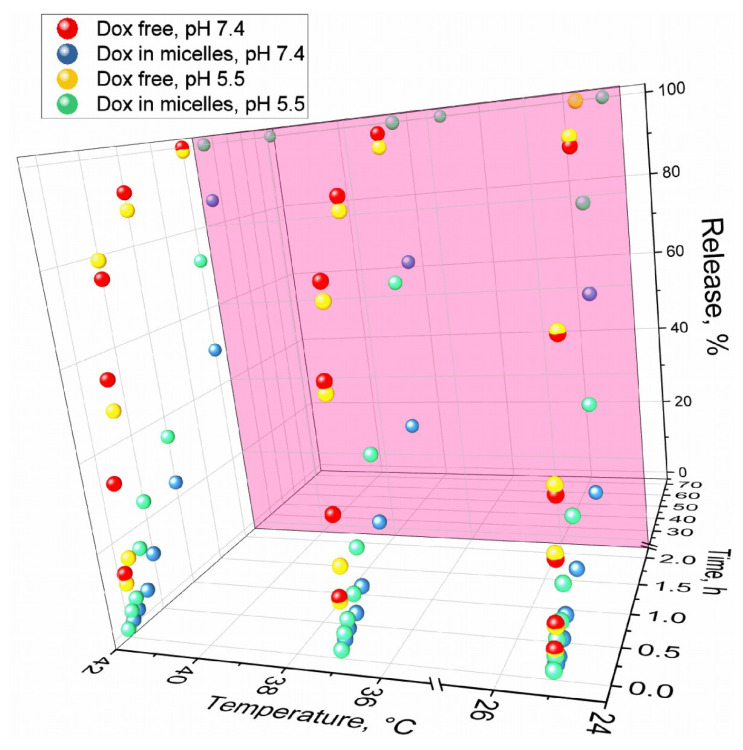
Release kinetics curves of free Dox and micellar (Chit5-MUA-20) forms of Dox complexes (1 mg/mL, 50% Dox *w*/*w*). Dialysis membrane (12–14 kDa cut-off) into an external solution (1:10 by volumes). PBS (pH = 7.4, 0.01 M), sodium acetate buffer (pH = 5.5, 0.01 M). Dox is detected by absorption at 488 nm. T = 25, 37, and 42 °C.

**Figure 7 pharmaceutics-15-01135-f007:**
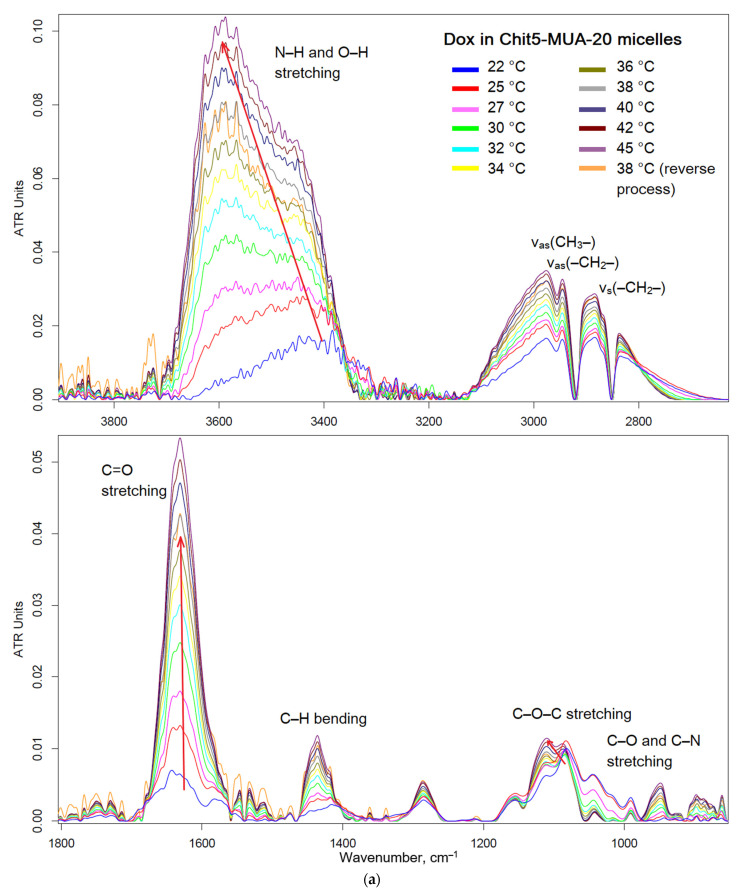
(**a**) FTIR spectra of Dox in Chit5-MUA-20 micelles at 22–45 °C. (**b**) Corresponding dependences of I1631, ratio I1080/I1115 (the peaks of chitosan C–O–C bonds, which characterize the ratio of the hydrophilic to hydrophobic component of chitosan), and 1630–1640 cm^−1^ peak position (the C=O bond, which characterizes the microenvironment of the carbonyl group of doxorubicin, which becomes more hydrophilic) on temperature. I—intensity.

**Figure 8 pharmaceutics-15-01135-f008:**
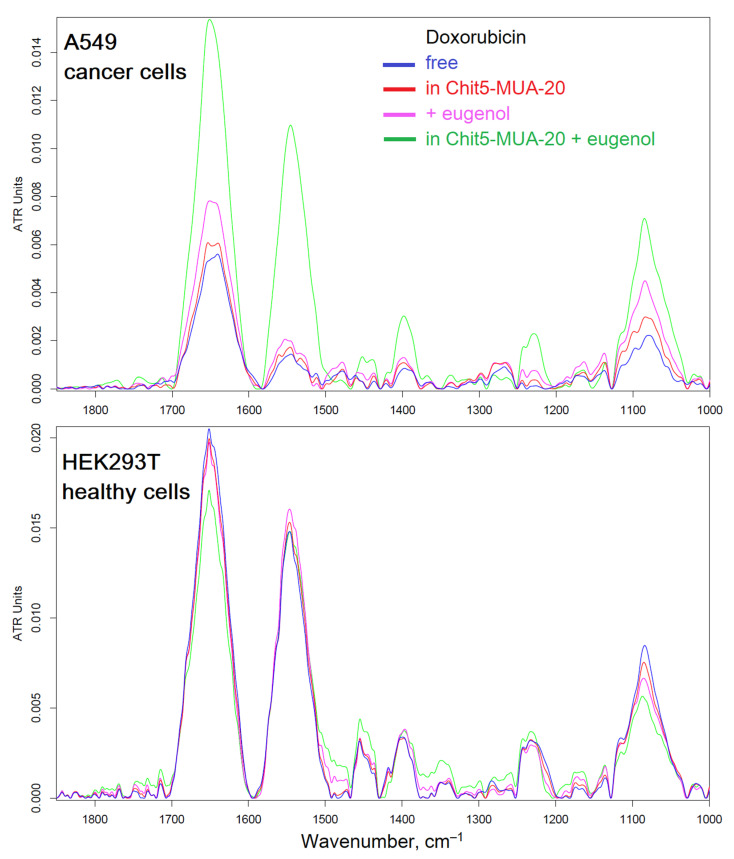
Difference FTIR spectra of cancer A549 cells and healthy HEK293T cells: after and before incubation for 3 h with doxorubicin (0.1 mg/mL). Free Dox, Dox + EG (0.1 mg/mL), or Dox in micellar form (polymer:Dox = 1:2 *w*/*w*) were used. pH = 7.4 (PBS, 0.01 M). T = 37 °C.

**Figure 9 pharmaceutics-15-01135-f009:**
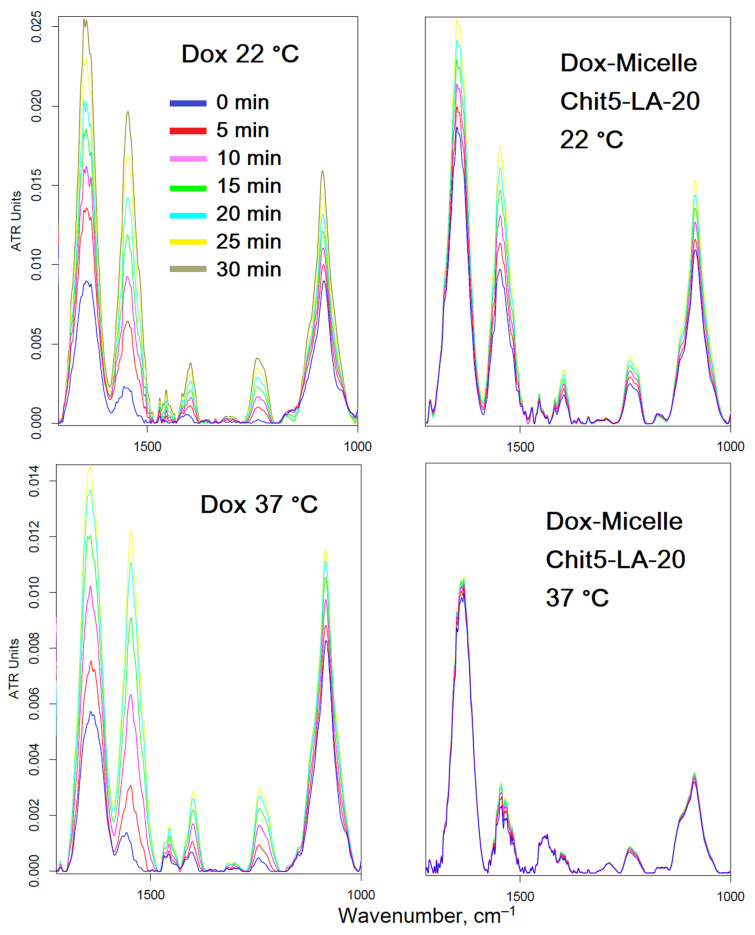
FTIR spectra of healthy HEK293T cells during incubation for 30 min with doxorubicin (0.1 mg/mL) in free form or in micellar form (polymer:Dox = 1:2 *w*/*w*). pH = 7.4 (PBS, 0.01 M). T = 22 or 37 °C.

**Figure 10 pharmaceutics-15-01135-f010:**
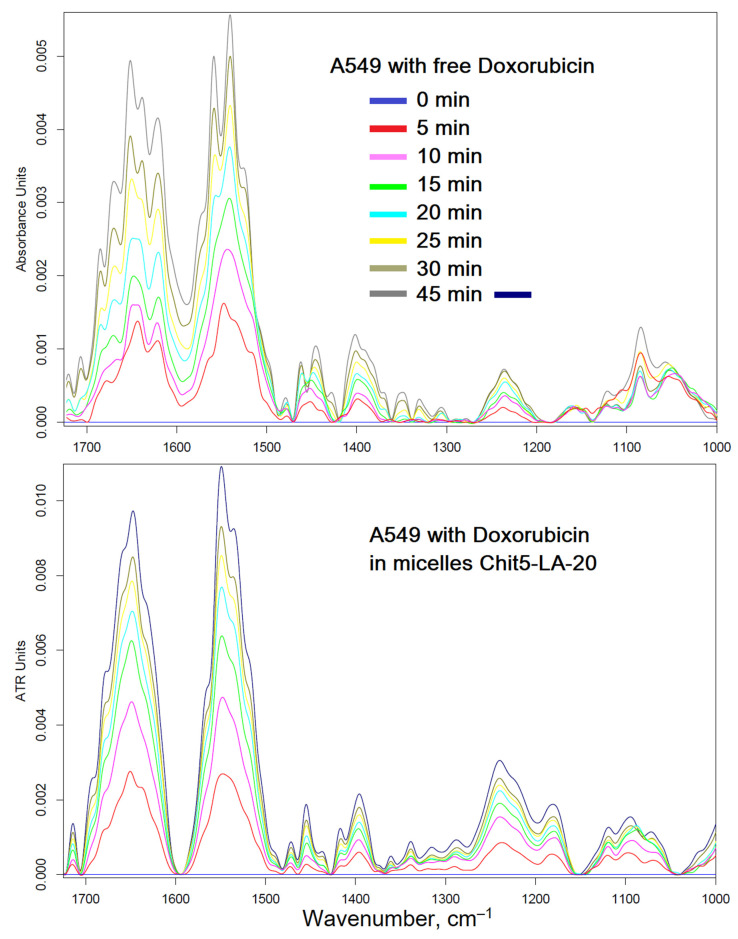
Difference FTIR spectra of cancer A549 cells during incubation for 45 min with doxorubicin (0.1 mg/mL) in free form or in micellar form (polymer:Dox = 1:2 *w*/*w*). pH = 7.4 (PBS, 0.01 M). T = 37 °C.

**Figure 11 pharmaceutics-15-01135-f011:**
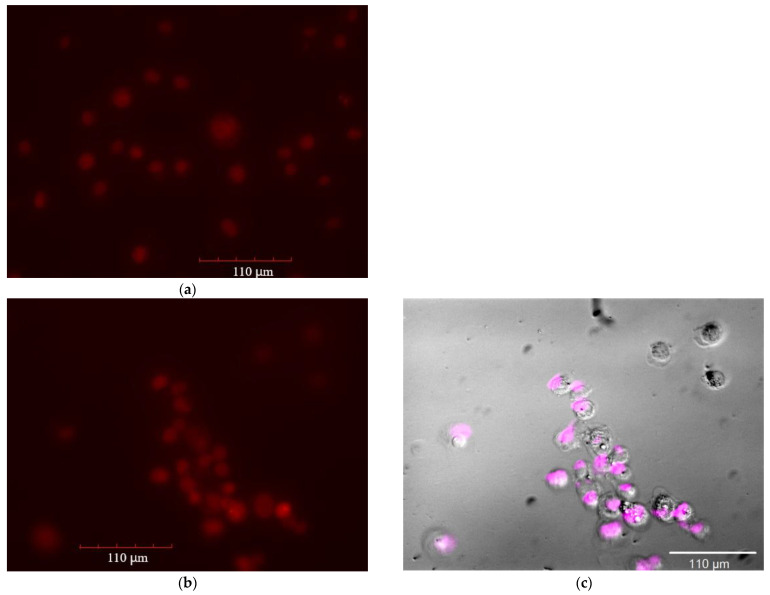
Fluorescence images of A549 after 2 h incubation with Dox 10 μg/mL: (**a**) free in Dox red channel; (**b**,**c**) in micelles Chit5-MUA-20 in Dox red channel and eosin magenta channel, respectively; (**c**) with EG; (**d**) in micelles Chit5-MUA-20 with EG in Dox red channel. λ_exci_ = 500–560 nm. (**e**) Fluorescence images of A549 after 2 h of incubation with Dox 10 μg/mL in eosin-labelled micelles Chit5-MUA-20: eosin (magenta) channel. λ_exci_ = 460–490 nm. The scale segment is 110 µm. (**f**,**g**) Optical images of A549 after 72 h incubation with Dox 1 mg/mL free and in micelles Chit5-MUA-20, respectively.

**Figure 12 pharmaceutics-15-01135-f012:**
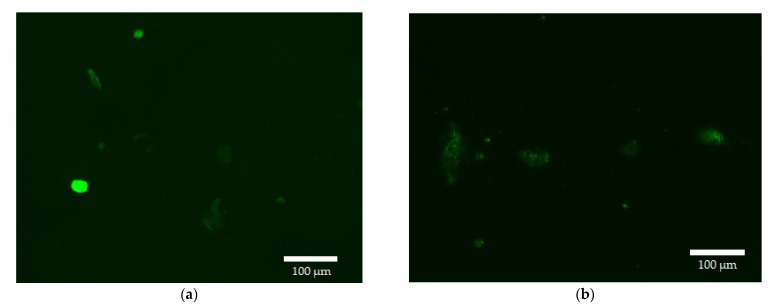
Fluorescence images of HEK293T after 2 h of incubation with Dox 10 μg/mL: (**a**) free; (**b**) in micelles of Chit5-MUA-20 with EG. Highlighted in green for ease of perception and so as not to confuse healthy cells with cancerous ones. The scale segment is 100 µm. λ_exci_ = 500–560 nm.

**Figure 13 pharmaceutics-15-01135-f013:**
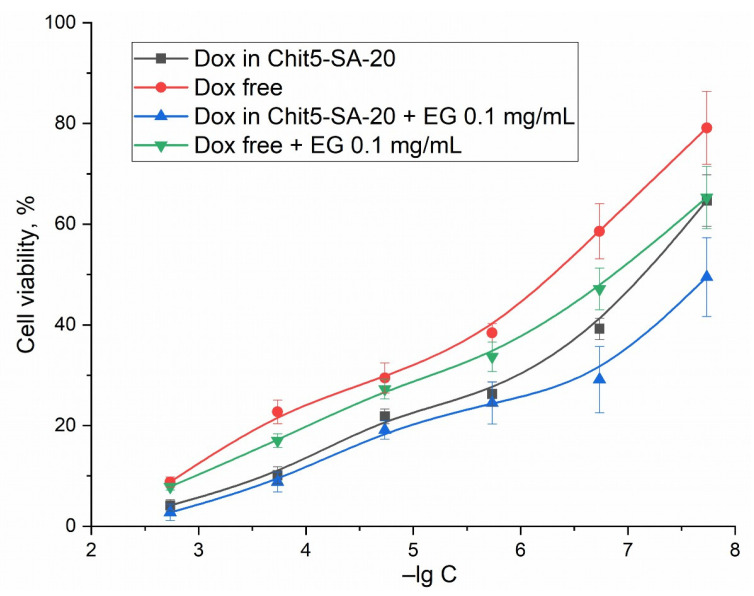
Dependences of A549 cells survival on the logarithm of concentrations of free Dox, Dox in micelles, and Dox enhanced with EG. RPMI-1640 medium supplemented with 5% fetal bovine serum and 1% sodium pyruvate at 5% CO_2_/95% air in a humidified atmosphere at 37 °C.

**Table 1 pharmaceutics-15-01135-t001:** Polymeric micelles’ physico-chemical characteristics.

Micelle Designation	Chitosan Modification Degree **, %	Molecular Weight of One Structure Unit, kDa	Hydrodynamic Diameter ***, nm	Critical Micelle Concentration, nM
Chit5-SA-20 *	12 ± 1	6.0 ± 0.3	116 ± 21	16 ± 4
Chit5-OA-20 *	20 ± 2	6.7 ± 0.4	95 ± 12	8 ± 2
Chit5-MUA-20 *	15 ± 2	6.0 ± 0.3	103 ± 17	40 ± 7
Chit5-LA-20 *	15 ± 2	5.9 ± 0.2	72 ± 13	7 ± 1

* Chit5-X-20, where 5 is molecular weight of chitosan in kDa and 20 is the theoretical degree of chitosan modification. X—acid residue; SA—stearic acid; OA—oleic acid; MUA—11-mercaptoundecanoic acid; LA—lipoic acid. ** The experimental modification degree was determined by TNBS spectrophotometric titration of NH_2_-groups of Chit5 and its conjugates and the FTIR peaks ratio. *** by Nanoparticle Tracking Analysis (NTA).

**Table 2 pharmaceutics-15-01135-t002:** Entrapment efficiency of Dox in polymeric micelles determined using FTIR spectra deconvolution with Lorentzian (Figure 5) and analytical dialysis (the data are given in parentheses). pH = 5.5 (sodium acetate buffer, 0.01 M), pH = 7.4 (PBS, 0.01 M). T = 37 °C.

Polymeric Micelle	Entrapment Efficiency of Dox, %	
	pH = 5.5	pH = 7.4
Chit5-SA-20	42 ± 3 (40 ± 4)	30 ± 2 (30 ± 4)
Chit5-OA-20	72 ± 5 (77 ± 3)	55 ± 5 (53 ± 6)
Chit5-MUA-20	60 ± 5 (63 ± 7)	46 ± 4 (48 ± 3)
Chit5-LA-20	66 ± 4 (65 ± 4)	58 ± 5 (53 ± 6)

**Table 3 pharmaceutics-15-01135-t003:** Initial rate of Dox release (%/h) at pH = 7.4 or 5.5 and T = 25, 37, or 42 °C. The conditions are similar to those given in Figure 6.

T, °C\pH	pH = 5.5	pH = 7.4
25 °C	22 ± 3	8 ± 1
37 °C	32 ± 4	15 ± 3
42 °C	49 ± 6	23 ± 5

**Table 4 pharmaceutics-15-01135-t004:** A549- and HEK293T-associated fluorescence (conventional units) depending on the composition of the Dox-containing formulation (10 μg/mL). Determined by fluorescent image analysis and fluorescence quantification of Dox uptake by the absorption of solution over cells. PBS (0.01 M, pH 7.4). T = 37 °C.

Formulation	A549-Associated Fluorescence	A549-Associated/Background Fluorescence Ratio	HEK293T-Associated Fluorescence
Dox free	71 ± 8	2.7 ± 0.5	93 ± 7
Dox in micelles Chit5-MUA-20	101 ± 14	3.5 ± 0.6	83 ± 5
Dox + eugenol	90 ± 11	10 ± 1	76 ± 9
Dox in micelles Chit5-MUA-20 + eugenol	175 ± 24	9± 1	53 ± 4

## Data Availability

The data presented in this study are available in the main text.

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
