# Peer review of "Smart pH- and Temperature-Sensitive Micelles Based on Chitosan Grafted with Fatty Acids to Increase the Efficiency and Selectivity of Doxorubicin and Its Adjuvant Regarding the Tumor Cells"

_pharmaceutics, 2023, doi:10.3390/pharmaceutics15041135_

Round 1

Reviewer 1 Report

This article reports the development of a polymer micellar system by grafting fatty acids with chitosan, which can encapsulate Dox and eugenol and retention of cytostatics in the tumor cells. The developed pH and temperature-sensitive polymer micelles have high entrapment efficiency for Dox and eugenol (EG), the entrapment efficiency is more than 60%, and can release drugs for a long time in a weakly acidic medium adapted to the tumor microenvironment. When Dox and EG adjuvant were used together, the penetration efficiency of Dox into cancer cells was increased by 2-3 times, but the permeability to HEK293T was reduced. In addition, the study is burdened with several shortcomings making it difficult to interpret the results, and there are many serious problems with this manuscript.

1. This article lacks the Transmission electron microscopy (TEM) pictures of the synthesized micelles, and the morphology of the micelles cannot be judged. It is recommended to supplement the TEM pictures of the micelles.

2. The FTIR diagram in Figure 2 is chaotic and the peak is not obvious. It is recommended to redraw it.

3. It is suggested to supplement the Hydrogen nuclear magnetic resonance1H NMR of Chit5 and different micelles.

4. Figure 4 fluorescence spectrum suggests showing a complete peak; Figure 4 and Figure 5 numbers in different positions, it is recommended to unify; Figure 4 (c) line thickness is not consistent with other drawings, horizontal and vertical font size is not consistent. It is recommended that all graphs be modified.

5. The meaning of the blue line is not marked in Figure 5, and Figure 5(b) has no free DOX for comparison.

6. About 60% of Chit5-MUA is released in PBS at pH 7.4 at 2 hours, but only 20% is released in PBS at pH 5.5. Whether it will be released in normal cells, but less in tumor cells, resulting in toxicity to normal cells?

7. The paper stated that polymer micelles have the characteristics of temperature-sensitive targeted drug release, but only the structural changes at different temperatures were provided. Please provide the data of targeted drug release at different temperatures.

8. The article mentions that micelles will not penetrate normal cells to release Dox at normal temperature, but it does not explain how micelles penetrate tumor cells to release drugs. Please give the information.

Reviewer 2 Report

1. In Fig. 1a, authors should put the name of the compound below its structure

2. In all figures, the labels (a, b, c...) should be put above the subfigures, not below them.

3. The captions of all figures are very confusing, authors should simplify them for better readability. 

4. In Fig. 1b, the authors should also provide bright field images along with fluorescent ones. 

5. In Fig. 2, Authors should provide the intact full spectrum of FTIR and then split them in the inset of zoom-in sections...

6. In Table 1., what is designation? authors should use appropriate words throughout the manuscript. 

7. In table 1., I think 'nanoparticle tracking analysis' will be in capital 'Nanoparticle Tracking Analysis (NTA)'

8. In Fig. 8, It should be 'Different' instead of 'Difference'

9. In Fig. 9 and others, authors should avoid putting text like 'Smart micelles act at 37C and protect healthy cells from cytostatics' rather mention it in the figure caption after formatting it properly. 

10. What does the extra line indicate at 45 min. in the top panel of fig. 10

11. In Fig. 11, Separate images should be arranged properly in a figure panel. Actually, authors should do this will all other necessary figures. 

Reviewer 3 Report

The paper submitted by Zlotnikov et al. deals with the synthesis and self-assembly of chitosan-based amphiphilic graft copolymers as delivery systems for DOX.

The aim of the study is interesting as the authors used FTIR for investigation of the interactions between the micellar system and the cells. Moreover, the conclusions are supported by the results. However, some corrections mandatory:

1. the introduction section must be completed with the following very recent reference: https://doi.org/10.3390/polym14214702

2. line 74: not micelles are amphiphilic but the copolymers

3.line 82: revise: "structure of healthy and cancer cells."

4.line 101: use the term "demicellization" instead of separation of polymeric micelles.

5.line 266: the authors must explain how the hydrogen bonds are typical for micellar systems. At which level these H-bonds appear in micelles?

6. line 268: revise "are shifted shift during..."

7. line 273: the acid's residues were conjugated to chitosan and not crosslinked.

8. table 1: how the "molecular weight of one structure unit" was calculated?

9. line 320, 470, 471: in a scientific paper the use of such types of expressions is not acceptable. Try not to use question and exclamation marks.

10. line 329: revise "1-2 orders of magnitude higher dye concentration."

11. line 335: what is the meaning of term "3-5 orders of magnitude better"

12. line 338: what is the difference between :compaction of the micelle core" and "compactization"?

13. add units in all figures (when necessary).

14. line 398: "revise "it quickly leaves". It's not scientific.

15. fig 11: which is the difference between photos 11 a, b, c and d? No significant difference is visible. The authors must discussed in detail the supposed difference.

16. 

Round 2

Reviewer 1 Report

No comments

Reviewer 2 Report

Authors have followed most of my comments.